# In Silico Analysis of Bioactive Peptides Released from Giant Grouper (*Epinephelus lanceolatus*) Roe Proteins Identified by Proteomics Approach

**DOI:** 10.3390/molecules23112910

**Published:** 2018-11-08

**Authors:** Fenny Crista A. Panjaitan, Honey Lyn R. Gomez, Yu-Wei Chang

**Affiliations:** 1Department of Food Science, National Taiwan Ocean University, Keelung 20224, Taiwan; fennycap@gmail.com; 2Institute of Fish Processing Technology, College of Fisheries and Ocean Sciences, University of the Philippines Visayas, Miagao, Iloilo 5023, Philippines; honeylyngomez23@yahoo.com

**Keywords:** Angiotensin-I converting enzyme (ACE-I), dipeptidyl peptidase-IV (DPP-IV), giant grouper roe, in silico, proteomic

## Abstract

Major proteins contained in dried giant grouper roe (GR) such as vitellogenin (from *Epinephelus coioides*; NCBI accession number: AAW29031.1), apolipoprotein A-1 precursor (from *Epinephelus coioides*; NCBI accession number: ACI01807.1) and apolipoprotein E (from *Epinephelus bruneus*; NCBI accession number: AEB31283.1) were characterized through compiled proteomics techniques (SDS-PAGE, in-gel digestion, mass spectrometry and on-line Mascot database analysis). These proteins were subjected to in silico analysis using BLAST and BIOPEP-UWM database. Sequence similarity search by BLAST revealed that the aligned vitellogenin sequences from *Epinephelus coioides* and *Epinephelus lanceolatus* share 70% identity, which indicates that the sequence sample has significant similarity with proteins in sequence databases. Moreover, prediction of potential bioactivities through BIOPEP-UWM database resulted in high numbers of peptides predominantly with dipeptidyl peptidase-IV (DPP-IV) and angiotensin-I-converting enzyme (ACE-I) inhibitory activities. Pepsin (pH > 2) was predicted to be the most promising enzyme for the production of bioactive peptides from GR protein, which theoretically released 82 DPP-IV inhibitory peptides and 47 ACE-I inhibitory peptides. Overall, this work highlighted the potentiality of giant grouper roe as raw material for the generation of pharmaceutical products. Furthermore, the application of proteomics and in silico techniques provided rapid identification of proteins and useful prediction of its potential bioactivities.

## 1. Introduction

Bioactive peptides such as antihypertensive [1,2,3] and anti-diabetic peptides [4,5] are widely observed due to their good potential as pharmaceutical products, particularly for human health enhancement. Hypertension is one of the major risk factors causing cardiovascular diseases, and generally occurs with obesity, pre-diabetes and atherosclerosis [6,7,8]. The degradation of angiotensin I and bradykinin by angiotensin-I converting enzyme (ACE-I) within the renin-angiotensin- aldosterone system (RAAS) stimulates the increase of blood pressure and leads to hypertension [9]. Followed by hypertension, diabetes mellitus type 2 (DM2) commonly occurs worldwide, and is one of the prime causes of death. Singh et al. [10] reported that dipeptidyl peptidase IV (DPP-IV) cleaves incretin hormones such as glucagon-like peptide 1 (GLP-1) and glucose-dependent insulinotropic peptide (GIP), resulting in DM2. Some synthetic therapeutic drugs have been developed to treat hypertension and DM2; however, the majority of them are considered unsafe due to the side effects associated with their consumption. These include inflammatory responses, taste disturbance, nausea, headache and allergic reaction [1,10,11]. Furthermore, bioactive peptides derived from natural sources like fish by-products [12,13,14], cereal grains [15,16] and dairy products [17,18] combined with drugs as supportive agents have been regarded as an alternative treatment for diseases. Additionally, the use of antihypertensive peptides in diet also contributes to prehypertensive treatment and/or stage 1 hypertension [19,20,21]. This is mainly owing to bioactive peptides with ACE-I and DDP-IV inhibitory activities.

Grouper is a family of fish species massively cultured in Taiwan, with a production volume of 22,930 metric tons in 2012 [22]. In terms of world production, the total wild catch and aquaculture production of grouper in 2009 were 242,136 metric tons and 74,722 metric tons, respectively [23]. Among the grouper species, giant grouper (*Epinephelus lanceolatus*) is popularly cultivated in Taiwan due to its fast-growing ability and high market price [24]. During spawning season, some of the fish roe fail to fertilize. The drowned, unfertilized roe are usually discarded, whereas the fertilized ones remain on the water surface. Nonetheless, unfertilized giant grouper roe can serve as potential raw materials for the production of bioactive peptides. Fish roe have been considered to be promising sources of protein, particularly tuna roe (18.44–20.15%) [25], Baltic herring roe and burbot roe (16.8% and 16.2%, respectively) [26], Pacific mackerel roe (25.3%) [27], red salmon roe (~26.8%) [28] and sea bream roe (19.2%) [29]. Thus, the utilization of unfertilized roe of giant grouper as a source of bioactive peptides, thereby increasing their value, appears feasible. In addition, not all fish roe are distributed to the market and sold at a high price. With regards to this, the idea of diversifying low-value and unfertilized roe to elevate roe product varieties like pharmaceutical drugs has been raised.

In the development of protein analysis, mass spectrometry (MS)-based proteomics has been popularly adopted and implemented to identify proteins and peptides from food materials. Proteomics is considered to be a new methodology that can analyze the entire proteins present in protein sample [30]. Once proteins are identified, the sequence can be further observed using in silico approaches, such as BIOPEP-UWM database [31,32] and BLAST [33]. The BIOPEP-UWM database, a database of biologically active peptide sequences documented from various resource materials, combined with proteomics, has been successfully used to investigate biological activities from food products like tilapia [34] and chickpea [35,36]. Moreover, BIOPEP-UWM can also carry out the simulation of enzymatic hydrolysis using certain proteases to estimate the release of the target bioactive peptides. The application of in silico tools will reduce the cost and time requirement with regard to the theoretical estimation of potential bioactivities and the corresponding activity of protein after hydrolysis using specific proteases. In addition, proteomics coupled with in silico techniques are able to deliver a rapid method for identification and characterization of huge numbers of proteins from complex food materials. Therefore, the objectives of this study are to identify the major proteins contained in giant grouper roe using a proteomics approach; to demonstrate in silico analysis for the prediction of ACE-I and DPP-IV inhibitory peptides encrypted in the protein sequence of giant grouper roe; and to screen proteases based on their capacity to release huge number of bioactive peptides through enzymatic hydrolysis simulation.

## 2. Results

### 2.1. Identified Proteins from Giant Grouper Roe

The crude protein content of GR was found to be 56.90 ± 0.28% (*w*/*w*, in dry weight basis). GR protein was characterized using SDS-PAGE to observe protein bands. Eight protein bands in 12% gel electrophoresis (Figure 1) were selected for further observation. All bands marked with rectangle boxes (labeled as A, B, C, D, E, F, G and H) were subjected to in-gel digestion and nano LC-ESI-MS/MS analysis.

The estimated molecular weights (MWs) of selected bands were 88.8 (A), 70.8 (B), 65.5 (C), 56.8 (D), 31.8 (E), 26.8 (F), 22.9 (G), and 10.1 (H) kDa. Table 1 shows the list of protein hits identified from giant grouper and their characteristics, including protein length, MWs from NCBI reference and estimated MWs from SDS-PAGE. The identified proteins were vitellogenin from *Epinephelus coioides* (NCBI accession number: AAW29031.1), apolipoprotein A-1 precursor (29.1 kDa) from *Epinephelus coioides* (NCBI accession number: ACI01807.1) and apolipoprotein E (30.7 kDa) from *Epinephelus bruneus* (NCBI accession number: AEB31283.1). As observed, some MWs of protein hits estimated from SDS-PAGE were quite comparable to the theoretical MWs presented in the NCBI database (accessed on 21 February 2018).

### 2.2. Identified Tryptic Peptide Sequences

Tryptic fragments matching with vitellogenin (NCBI accession number: AAW29031.1) from band B were examined by nano LC-ESI-MS/MS analysis to determine fragments relevant to protein hits obtained from Mascot ion search results (accessed on 21 February 2018) as shown in Appendix A. Tryptic peptides were observed to have doubly and triply charged signals. Figure 2A shows the LC-ESI-MS spectrum of LVEPELFEYSGVYPK (MW 1768.89) detected from vitellogenin in band B, illustrating the doubly charged peptide with 885.45 *m*/*z*. In addition, insert 1 displays the double charge signal distinguished by the 0.5 difference between adjacent signals while insert 2 shows the fragmentation of the spectrum of the identified peptide.

Similarly, the peptide YEFSTELLQTPIQLIK (MW 1922.04) derived from the same band represents the triply charged peptide with 641.68 *m*/*z*, as shown in Figure 2B. The triple charge signal with a difference of 0.33 between adjacent signals is presented in insert 1, while the fragmentation of the peptide spectrum is illustrated in insert 2.

### 2.3. Homology Study of Vitellogenin

Vitellogenin sequences (NCBI accession number: AAW29031.1) from *Epinephelus coioides* were compared to those of *Epinephelus lanceolatus*. Research conducted by Om et al. [37] successfully identified vitellogenin sequences from giant grouper (*Epinephelus lanceolatus*) (as shown in Appendix A) using combined application of proteomics and Lutefisk, a computer-based sequencing algorithm. The homology study of two protein sequences was then performed by BLAST analysis shown in Figure 3. 

Line X represents vitellogenin sequences (NCBI accession number: AAW29031.1) of *Epinephelus coioides* (AAs positions from 2 to 615); line Y shows AA sequences of vitellogenin from *Epinephelus lanceolatus* (AAs positions from 35 to 648); and data between line X and Y illustrates the similarity of the AAs from two vitellogenin sequences. BLAST analysis generates a data of the “identities”, “positives” and “gaps” of two or more aligned AA sequences. The identities and the positives of the two AA sequences used in this study were 429/614 (70%) and 520/614 (84%), respectively, with a gap of 0/614 (0%). Also, the AA sequences of vitellogenin from *Epinephelus coioides* can also be present in *Epinephelus lanceolatus*, since the identity value (70%) was observed to be comparatively high.

### 2.4. Prediction of Potential Bioactive Peptides

Identification of protein bioactivities from giant grouper roe was demonstrated using BIOPEP-UWM database (accessed on 30 October 2018). AA sequences analyzed in the BIOPEP-UWM were vitellogenin (NCBI accession number: AAW29031.1) and apolipoprotein A-1 precursor (NCBI accession number: ACI01807.1), originated from *Epinephelus coioides*, and apolipoprotein E (NCBI accession number: AEB31283.1) of *Epinephelus bruneus* origin. The results from the BIOPEP-UWM database for those proteins are shown completely in Appendix A. Profiles of potential biological activities from the protein hits and their frequency of occurrence (A) within the protein sequences are displayed in Table 2. Common bioactivities detected from the protein fragments were ACE-I and DPP-IV inhibitory activities. The A value of protein hits within the intact protein sequences ranged from 0.334–0.408 and 0.589–0.687, respectively. Minkiewicz et al. [38] mentioned that the occurrence of bioactive fragments (A) generated in protein sequences is related to the AA composition contributing to its bioactivities.

Vitellogenin (NCBI accession number: AAW29031.1) was selected for bioactive peptide profiling, as revealed in Figure 4. Result shows that vitellogenin generated huge number of DPP-IV inhibitory peptides with 423 fragments (marked with green lines), 251 ACE-I inhibitory peptides (marked with blue lines), 49 antioxidant inhibitory peptides (marked with yellow lines) and some other peptides; immunomodulating, stimulating, immunostimulating, regulating, neuropeptide, bacteria permease ligand, inhibitor, hypotensive, activating and ubiquitin-mediated proteolysis (marked with black lines).

### 2.5. Potential Bioactive Peptides by In Silico Cleavage Analysis

The BIOPEP “enzyme action” tool was used to perform proteolysis simulation of all the available proteases in the database. During the simulation, some proteases were not able to release bioactive peptides optimally. Nevertheless, many proteases exhibited high potential in generating a large number of peptides with bioactivities, predominantly DPP-IV inhibitory peptides (Table 3). Thus, the result is comparable to the predicted bioactive peptides reflected in Table 2.

Overall, pepsin (pH > 2) generated a significant number of DPP-IV and ACE-I inhibitory fragments (82 and 47, respectively). Other single-action enzymes such as bromelain, chymotrypsin A, chymotrypsin C, ficin, pancreatic elastase, papain, proteinase K and thermolysin also released relatively high numbers of bioactive peptides in computational analysis. The “enzyme action” tool in the BIOPEP-UWM database can also be utilized on combined enzymes (maximum 3 proteases) to simulate small intestinal (trypsin and chymotrypsin A; trypsin and chymotrypsin C) and gastric (pepsin, trypsin and chymotrypsin A; pepsin, trypsin and chymotrypsin C) digestion. The results of the simulation show that the integration of three enzymes (pepsin, trypsin and chymotrypsin A; pepsin, trypsin and chymotrypsin C) led to better production of bioactive peptides compared to the two enzymes trypsin and chymotrypsin A, as well as trypsin and chymotrypsin C.

## 3. Discussion

In this study, vitellogenin (from *Epinephelus coioides* NCBI accession number: AAW29031.1) was detected in all marked bands in the gel. Being the major protein in fish roe, the possibility of its occurrence is expected to be high. Vitellogenin is cleaved into several fragments, such as lipovitellin (Lv), phosvitin (Pv) and β′-component (β′-c), during the maturation period of oocytes and then stored in the egg yolk [39]. These proteins have different MWs; Lv consists two heavy-chains and two light-chains (appearing at 92 kDa and 29 kDa, respectively), phosvitin migrates around 6–23 kDa and β′-c protein (34 kDa) is composed of 17 kDa fractions [40,41]. Among all the bands corresponding to different MWs of vitellogenin, band B was selected because its MW (70.8 kDa) is reasonably comparable to the reference MW from the NCBI database (67.9 kDa). In addition, it also showed a high percentage of sequence coverage, at 41%. The percentage of sequence coverage from the on-line Mascot database revealed the sequence similarity of the identified tryptic peptides to corresponding protein hits. Other protein hits were also identified from band E (31.8 kDa) and band F (26.8 kDa), such as apolipoprotein A-1 precursor (29.1 kDa) from *Epinephelus coioides* (NCBI accession number: ACI01807.1) and apolipoprotein E (30.7 kDa) from *Epinephelus bruneus* (NCBI accession number: AEB31283.1). It was reviewed by Babin and Vernier [42] that apolipoprotein is the structural component of lipoprotein tending to bind lipid during oocytes development. In previous studies, apolipoprotein was characterized using SDS-PAGE and the position of apolipoprotein A-1 (apoA-I-like) and apolipoprotein E (apo E) were presented at ~25–28 kDa and ~34 kDa, respectively [43,44].

According to the Mascot database, none of the identified proteins originated from giant grouper (*Epinephelus lanceolatus*). As observed, the NCBI database (accessed on 21 February 2018) provided only 312 proteins of *Epinephelus lanceolatus*, which were mostly enzymes for oxidative phosphorylation such as cytochrome oxidase, nicotinamide adenine dinucleotide (NADH) dehydrogenase and adenosine triphosphate (ATP) synthase. In other words, information about protein sequences stored in the NCBI database is still limited, especially for major or storage proteins from giant grouper roe. In addition, protein hits from *Epinephelus coioides* and *Epinephelus bruneus* were selected in this study, since they are classified in the same genus with *Epinephelus lanceolatus*.

BLAST analysis was applied to compare the two vitellogenin sequences from *Epinephelus coioides* (NCBI accession number: ACI01807.1) and *Epinephelus lanceolatus* (identified by Om et al. [37]) and to calculate regions of commonality of AA sequences within protein [33]. Through blast analysis, it was found that vitellogenin from *Epinephelus coioides* (NCBI accession number: ACI01807.1) and *Epinephelus lanceolatus* identified by Om, et al. [37] scored 70% for identities, 84% for positives and 0% for gaps (Figure 4). Altschul et al. [33] mentioned that the “identities” represent the percentage of matched AAs within the total length of the aligned sequences, which means that the higher the identity, the more homologous they are; the “positives” characterize the percentage of matched AAs represented by the substitutions “+” within the total length of the sequences; and the “gaps” refer to the number of “−” within the total length of the AAs. The same biological classification of *Epinephelus coioides* and *Epinephelus lanceolatus* gave high homology in terms of AA sequences. Moreover, vitellogenin was detected in all bands, appearing in the SDS-PAGE gel. Therefore, these vitellogenin sequences from *Epinephelus coioides* originally detected in GR protein can probably be used as representatives of vitellogenin sequences from giant grouper roe for further observation of potential activities using BIOPEP-UWM database, as revealed in Figure 2.

A total of 423 DPP-IV inhibitory peptides were released from vitellogenin, and most of their sequences contained A (alanine) and P (proline) residues, which contributed to DPP-IV inhibitory activities. Davy et al. [45] claimed that P, A, S (serine) and Hyp (hydroxyproline) located on the second N-terminal position in sequences, for example, Xaa-P were capable of inhibiting DPP-IV activity. Most of the sequences found in this study were described in dipeptides and tripeptides, where some of them have overlapping bioactivities and possessed more than one activity (marked with red lines). One example of these is the KF, which has antihypertensive, inhibitor, hypotensive and DPP-IV inhibitory activities. According to quantitative structure–activity relationship (QSAR) study, AA compositions containing hydrophobic AA have great roles in the inhibition of DPP-IV and ACE-I activities [46,47,48]. In a previous study, Huang et al. [34] mentioned that tilapia skin and frame protein isolates theoretically produced abundant ACE-I inhibitory peptides based on BIOPEP-UWM database analysis. It was then confirmed by Lin et al. [49] that tilapia frame protein isolates hydrolyzed by pepsin can generate strong antihypertensive effects through in vitro (IC_50_: 0.57 mg/mL). Thus, the utilization of the BIOPEP-UWM database has been successfully applied for the prediction of potential bioactive peptides and gives propitious results with regards to determination of bioactivities from food protein sources. However, results obtained from the BIOPEP-UWM database might change, because data are continuously updated and revised with new sequences of both proteins and peptides. Minkiewicz et al. [31] stated that data concerning 659 proteins and 1968 peptides with 37 types of bioactivity were gathered in BIOPEP. At present, there are 740 proteins and 3695 peptides, with 45 types of bioactivity, that have been registered (accessed on 30 October 2018).

The BIOPEP-UWM database can also reveal which enzyme/s are able to release greater bioactive peptides with the use of the “enzyme action” tool. The application of this tool will provide efficient and effective screening of proteases that theoretically cleave proteins and release bioactive peptides. Pepsin (pH > 2) was predicted to release a greater number of preferable bioactive peptides compared to other single-action enzymes. Dunn [50] mentioned that pepsin preferred to cleave peptide bonds between hydrophobic AAs, for example, A and P. Apparently, pepsin is capable of releasing peptides with biological activities, since these peptides might have at least one hydrophobic AA in N-terminal residue. From this perspective, the use of BIOPEP “enzyme action” tool can be considered a rapid method of screening proteases and can be used as an alternative for random selection of enzymes through in vitro hydrolysis. However, it should be noted that the bioactive peptides theoretically produced by proteases may not always be comparable to those actually generated through in vitro analysis, due to some factors. In silico hydrolysis works according to the assumption that all cleavable peptide bonds within a certain protein will be accessible to the enzyme and will be easily hydrolyzed [51]. However, there is no guarantee that the situation will be same in in vitro hydrolysis, since the cleavage of peptide bonds during enzymatic hydrolysis depends on their accessibility and enzyme activity. In addition, the complexity of the protein structure might hinder protease–protein interactions, which could result in a mismatch of predicted and experimentally released peptides [52]. Previous studies have reported discordances between in silico and in vitro result, for example papain-catalyzed hydrolysis of bovine collagen [53] and pepsin hydrolysis result of potato proteins [54]. Therefore, bioactive peptides identified through in silico studies still need to be investigated further by in vitro analysis.

## 4. Materials and Methods

### 4.1. Materials

Frozen giant grouper roe were provided by Long Diann Marine Biotechnology Co., Ltd., Pingtung, Taiwan for research purposes. They were transported to the laboratory and stored at −20 °C until use. All chemicals used in this research were of analytical grade.

### 4.2. Preparation of Dried Giant Grouper Roe (GR)

The preparation of dried giant grouper roe (GR) was based on the method of Chalamaiah et al. [55], with some modifications. Frozen giant grouper roe were thawed at 4 °C and subjected to homogenization with deionized water (1:2, *w*/*v*) using high-speed blender. The resulting homogenate was then lyophilized and stored at −20 °C for further experimental works. The protein content was calculated using the Kjeldahl method [56]; a conversion factor of 6.25 was used to convert the nitrogen content to protein content.

### 4.3. Sodium Dodecyl Sulphate-Polyacrylamide Gel Electrophoresis (SDS-PAGE)

Protein patterns of GR were measured using SDS-PAGE as described by Laemmli [57] with 4% stacking gel and 12% resolving gel. GR (4 mg) were dissolved in 1000 µL of sample buffer (0.5 M Tris-HCl pH 6.8, glycerol, 10% SDS, 0.5% bromophenol blue, *w*/*v*) and heated at 95 °C for 4 min. Afterwards, 5 µL of standard protein ladder (AccuRuler RGB prestained protein ladder, MaestroGen Inc., Hsinchu City, Taiwan) and 10 µL of sample solution were loaded into each well. The standard protein ladder contained ten proteins of 10, 17, 28, 35, 48, 63, 75, 100, 130 and 180 kDa as basis for molecular weight (MW) estimation. The voltage of power supply was set at 70 V for 30 min in stacking gel and 110 V for 90 min in resolving gel. The gel was stained with Coomassie Brilliant Blue R-250 for 30 min, and then soaked in destaining solution (water: methanol: acetic acid, 7:2:1, *v*/*v*/*v*) thrice for 15 min while being shaken by an orbital shaker (Fristek S10, Taichung City, Taiwan). After that, the gel was scanned with an E-Box VX5 device (Vilber Lourmat, Paris, France) and the captured image was analyzed using Vision-Capt software (V16.08a, Vilber Lourmat, Paris, France) to estimate the molecular weight of each band.

### 4.4. Trypsin In-Gel Digestion

The protocol was adapted from the method used by Shevchenko et al. [58], with several modifications. Selected protein bands of SDS-PAGE gel were excised from the gel and cut into 1 mm^3^ pieces. The gel pieces were placed into 650 µL microcentrifuge PP tubes (siliconized, methanol washed) and destained completely with 50% acetonitrile (ACN)/25 mM ammonium bicarbonate (ABC) solution. Destained gel pieces were then soaked in 100 µL of 50 mM dithioerythreitol (DTE)/25 mM ABC for 1 h at 37 °C. After which, the mixture was centrifuged and the DTE solution was discarded. The gel pieces were then subjected to alkylation process by incubating it with 100 µL of 100 mM iodoacetamide (IAM)/25 mM ABC for 1 h in the dark at room temperature. After the centrifugation, the IAM solution was removed and the remaining gel pieces were washed with 200 µL of 50% ACN/25 mM ABC for 15 min and the supernatant was discarded after centrifugation. The washing step was repeated 4 times. The gel pieces were then soaked with 100 µL of 100% ACN for 5 min and dried for about 5 min using Speed Vac (Thermo Scientific, Waltham, MA, USA). After that, the trypsin digestion proceeded by adding Lys-C/25 mM ABC (enzyme: protein, 1:50) and was incubated at 37 °C for 3 h. The gel mixture was combined with the same amount of trypsin and reincubated for at least 16 h at the same temperature. The tryptic peptides were extracted twice with 50 µL of 50% ACN/5% trifluoroacetic acid (TFA), and the resulting extract was transferred into a microcentrifuge tube. The combined extract was then dried in a Speed Vac device and purified through zip-tip purification process and the final sample peptides were kept for further nano LC-ESI-MS/MS analysis.

### 4.5. Nano LC-ESI-MS/MS Analysis

Tryptic peptide extracts were subjected to nano LC-ESI-MS/MS analysis using a nanoAcquity system (Waters, Milford, MA, USA). The unit was connected to the orbitrap elite hybrid mass spectrometer (Thermo Electron, Bremen, Germany) equipped with a PicoView nanospray interface (New Objective, Woburn, MA, USA). The dried tryptic peptides were suspended in 40 µL mobile phase solution (water with 1% formic acid/acetonitrile, 98:2, *v*/*v*) and centrifuged with 15,000× *g* speed at 4 °C for 15 min (3500, Kubota, Japan). An aliquot of peptide mixtures (20 µL) was loaded into the sample vial. LC separation was done using a C_18_ BEH column (Waters, Milford, MA, USA) and a segment gradient mobile phase consisting of solvent A (0.1% formic acid in water; *v*/*v*) and solvent B (acetonitrile with 0.1% formic acid, *v*/*v*). Solvent B increased from 5% to 35% for 60 min at a flow rate of 300 nL/min and a column temperature of 35 °C. Mass spectrometry was operated in the dara-dependent mode and the MS spectra were acquired in the orbitrap (*m/z* 350–1600) with a resolution of 120 K at 400 *m*/*z* and automatic gain control (AGC) target at 10^8^. The 20 most intense ions were sequentially isolated for CID MS/MS fragmentation and detection in the linear ion trap (AGC target at 10,000) with previously selected ions dynamically excluded for 60 s. Ions with single and unrecognized charge state were excluded.

### 4.6. MS/MS Data Analysis of Protein and Peptide Identification

MS/MS raw data were converted to PKL files before being subjected to a Mascot database search (http://www.matrixscience.com/search_form_select.html). The data were searched against the National Center for Biotechnology Information (NCBI) database (https://www.ncbi.nlm.nih.gov/) for all entries. Search parameters used were Actinoptergii (ray-finned fishes) for taxonomy entry, carbamidomethyl (C) and oxidation (M) as variable modification, 10 ppm peptide mass tolerance, 2+, 3+, 4+ peptide charge, ±0.6 Da MS/MS tolerance, ESI-TRAP as instrument, and trypsin as enzyme set with 2 missed cleavages. All peptide masses were obtained as monoisotopic masses.

In score distribution of peptide, the ion score was −10 log (P), where P is the probability that the observed match is a random event. Protein scores were attained from ion scores as a non-probabilistic basis for ranking protein families. Mascot search results showed the protein sequence coverage in percentage (%) expressing the sequence homology of identified tryptic peptides from giant grouper roe fractions to protein hits.

### 4.7. Basic Local Alignment Search Tool (BLAST) Analysis of Vitellogenin Sequences

Vitellogenin (NCBI accession number: AAW29031.1) from *Epinephelus coioides* obtained from the Mascot ion search was aligned with that of *Epinephelus lanceolatus* reported by Om et al. [37]. The homology study was performed using BLAST (https://blast.ncbi.nlm.nih.gov/Blast.cgi) and the scores, identities (%), positives (%) and gaps (%) between two protein sequences were determined.

### 4.8. BIOPEP-UWM Database Analysis

Giant grouper roe proteins identified by MS/MS data analysis were further investigated using BIOPEP-UWM database (http://www.uwm.edu.pl/biochemia/index.php/en/biopep) to predict the potential biological activities and to screen proteolytic enzymes based on their capacity to produce potential peptides. Firstly, “Bioactive peptides” was chosen from the “database” options. The identified proteins from giant grouper roe were analyzed using the “profiles of potential biological activity” tool, and the BIOPEP ID, name of peptide, activity, number of peptide, sequence and location of bioactive peptides in protein sequences were acquired. The occurrence of frequency (A) of bioactive peptides was calculated as A = a/N, where a = number of bioactive peptides and N = total number of amino acid (AA) residues in the protein chain [38].

Thereafter, the “enzyme(s) action” tool was chosen to simulate the enzymatic hydrolysis by using single-action proteinases and combined-action proteinases (maximum of three enzymes). After that, the “search for active fragments” option was selected to view the list of peptides containing potential activities.

## 5. Conclusions

The combination of proteomics and in silico analysis is a rapid method for identification of proteins from giant grouper roe, prediction of bioactive peptides possessed by identified protein sequences, and determination of appropriate protease/s that can theoretically release more bioactive peptides. Three proteins corresponding to identified tryptic peptides of giant grouper roe were discovered, viz., vitellogenin, apolipoprotein A-1 and apolipoprotein E. Sequence similarity search by BLAST revealed that the aligned vitellogenin sequences from *Epinephelus coioides* and *Epinephelus lanceolatus* shared 70% identity. This indicates that the sequence sample has significant similarity with proteins in sequence databases. With BIOPEP analysis, numerous ACE-I and DPP-IV inhibitory peptides were produced from intact proteins. Pepsin (pH > 2) and mixed proteases (pepsin, trypsin and chymotrypsin A; pepsin, trypsin and chymotrypsin C) exhibited better production of bioactive fragments from vitellogenin protein compared to other proteases. According to the reported data, giant grouper roe can be considered a potential and promising raw material for the generation of desirable bioactive peptides, especially with ACE-I and DPP-IV inhibitory activities. Moreover, they can be developed as pharmaceutical products to increase their market value and to provide enhancement to human health.

## Figures and Tables

**Figure 1 molecules-23-02910-f001:**
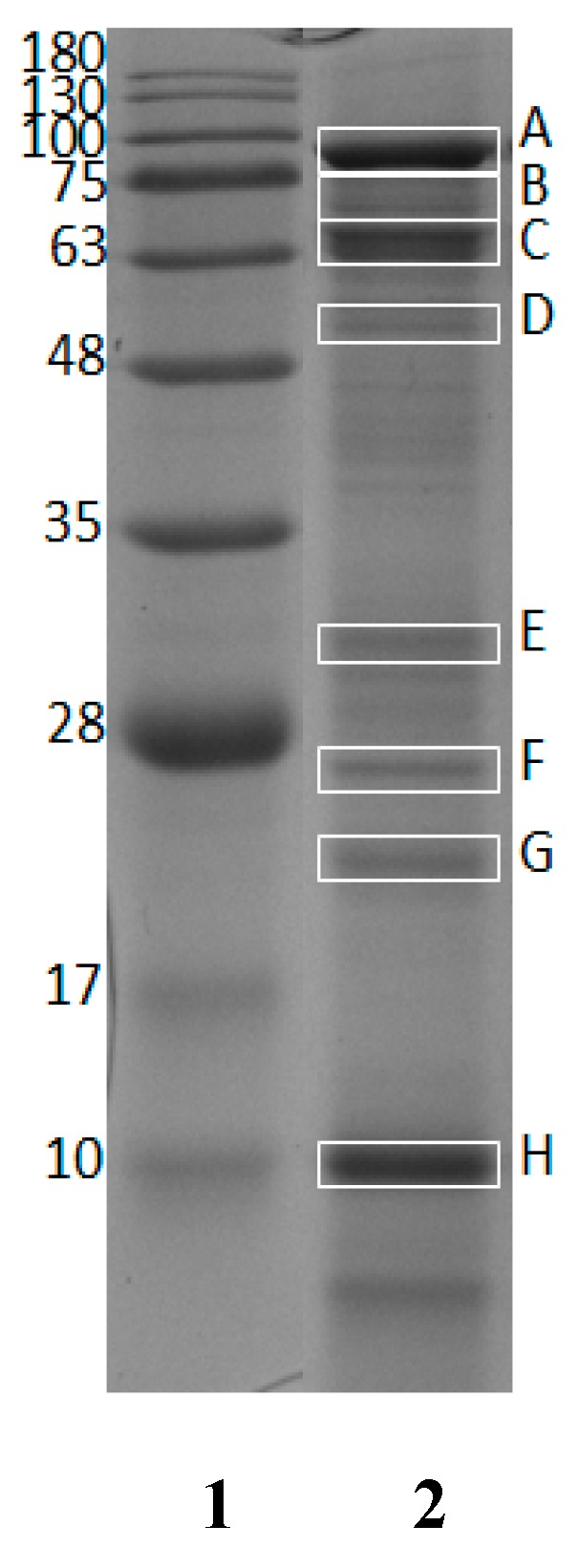
Twelve percent SDS-PAGE of dried giant grouper roe. Lane **1**: standard protein markers; Lane **2**: dried giant grouper roe.

**Figure 2 molecules-23-02910-f002:**
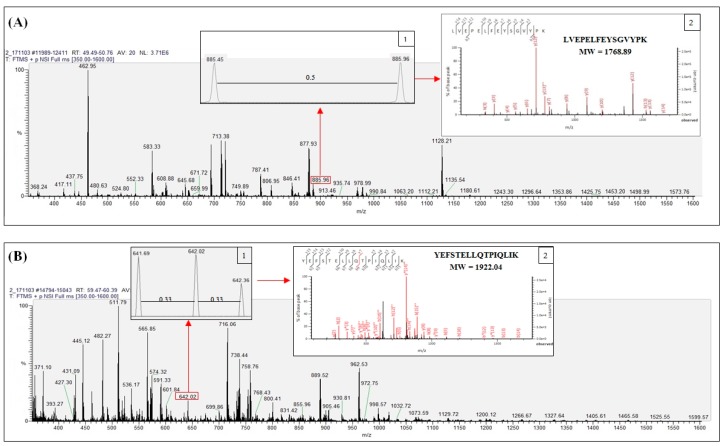
Nano LC-ESI-MS/MS spectrum scanning at *m*/*z* region 350 to 1600 Da from giant grouper roe protein band B with representative spectrum of identified tryptic peptides in doubly (**A**) and triply (**B**) charged signal.

**Figure 3 molecules-23-02910-f003:**
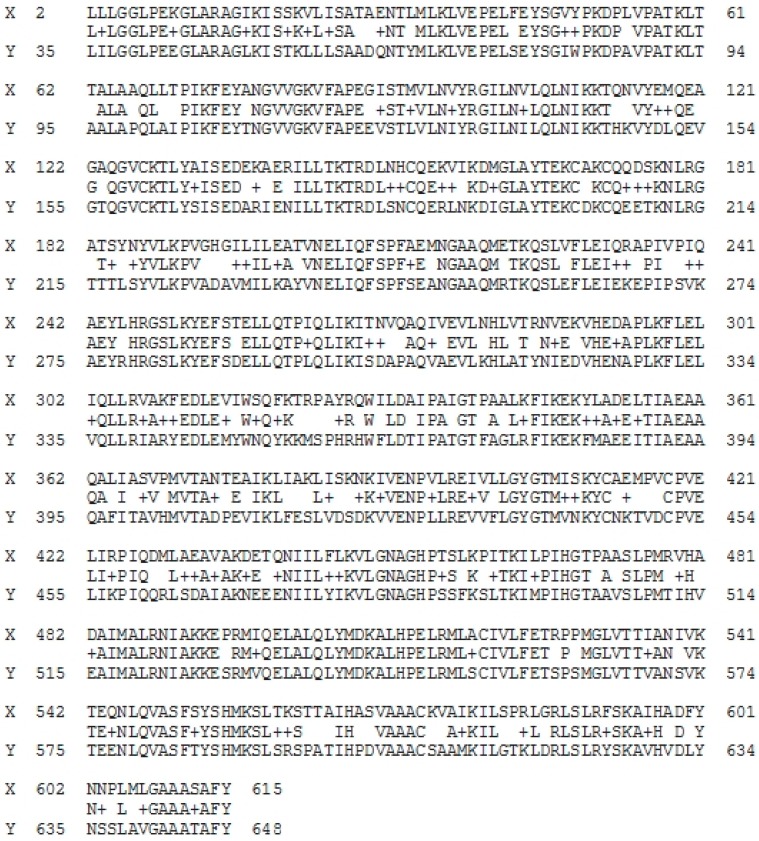
Blast analysis of aligned vitellogenin sequences from *Epinephelus coioides* (NCBI accession number: AAW29031.1; line X) and *Epinephelus lanceolatus* (obtained from Om et al. (2013); line Y) (accessed on 12 March 2018).

**Figure 4 molecules-23-02910-f004:**
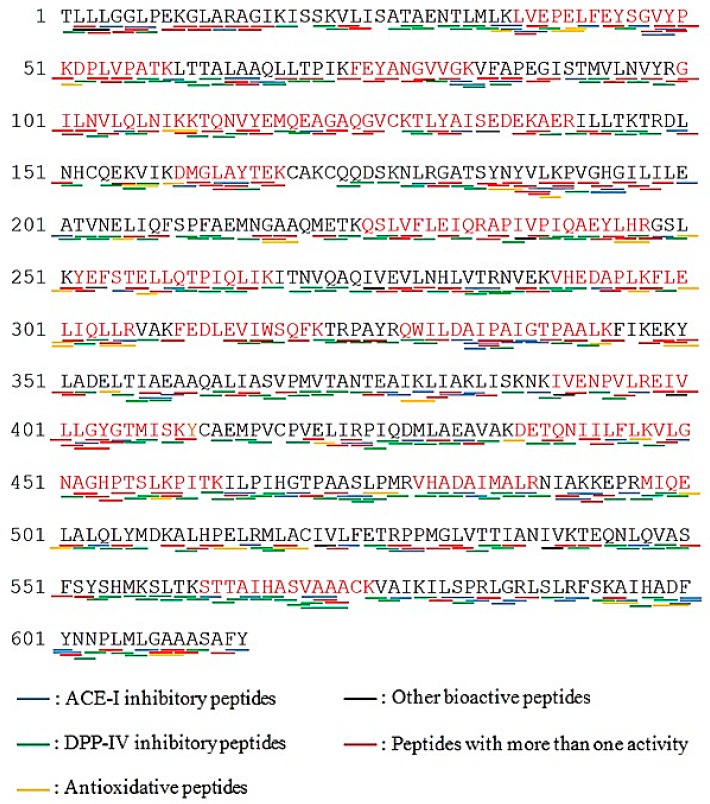
Protein sequences of vitellogenin (originated from *Epinephelus coioides*; NCBI accession number: AAW29031.1; Protein sequence coverage: 41%; 615 aa; excised from band B). Matching tryptic fragments were shown in red letters. Underlines presented bioactive peptides profiling.

**Table 1 molecules-23-02910-t001:** List of identified proteins from giant grouper roe and their characteristics.

Protein Name	Accession Number (NCBI)/Species ^a^	Protein Length ^a^	Score ^a^	Sequence Coverage (%) ^a^	MW (kDa) in NCBI Database ^a^	MW (kDa) in SDS-Page *
Vitellogenin	AAW29031.1/ *Epinephelus coioides*	615 aa	544	41	67.9	88.8 (A), 70.8 (B), 65.8 (C), 56.8 (D), 31.8 (E), 26.8 (F), 22.9 (G), 10.1 (H)
Apolipoprotein A-1 precursor	ACI01807.1/ *Epinephelus coioides*	263 aa	173	19	29.1	31.8 (E), 26.8 (F)
Apolipoprotein E	AEB31283.1/ *Epinephelus bruneus*	275 aa	629	42	30.7	31.8 (E), 26.8 (F)

* Molecular weight (kDa) in SDS-PAGE was determined using VisionCapt software. ^a^ Information was accessed from mascot ion search result on 21 February 2018.

**Table 2 molecules-23-02910-t002:** Prediction of potential bioactive peptides of giant grouper roe protein by BIOPEP-UWM database (accessed on 30 October 2018).

Protein Name	Number of Peptides			
	DPP-IV Inhibitory Activity	ACE Inhibitory Activity	Antioxidant Activity	Other Activities *
Vitellogenin (AAW29031.1)	423 (0.687)	251 (0.408)	49 (0.079)	87
Apolipoprotein A-1 precursor (ACI01807.1)	172 (0.654)	102 (0.388)	16 (0.061)	24
Apolipoprotein E (AEB31283.1)	162 (0.589)	94 (0.334)	16 (0.058)	22

Numbers in parentheses represent the frequency of occurrence (A) of bioactive peptides rounded off to 3rd decimal place. * Other activities: antibacterial, immunomodulating, stimulating, neuropeptide, regulating, inhibitor, hypotensive, activating ubiquitin-mediated proteolysis.

**Table 3 molecules-23-02910-t003:** Prediction of potential bioactive hydrolysates of vitellogenin protein (Accession number: AAW29031.1) using “enzyme action” tool (accessed on 30 October 2018).

Protease (EC Number)	Vitellogenin			
	DPP-IV Inhibition	ACE Inhibition	Antioxidant	Other Activities *
Bromelain (EC 3.4.22.32)	58	34	5	15
Calpain 2 (EC 3.4.22.53)	61	29	4	13
Cathepsin G (EC 3.4.21.20)	26	15	4	9
Chymase (EC 3.4.21.39)	20	14	2	9
Chymosin (EC 3.4.23.4)	ND	ND	ND	ND
Chymotrypsin A (EC 3.4.21.1)	30	18	7	11
Chymotrypsin C (EC 3.4.21.2)	52	28	7	7
Clostripain (EC 3.4.22.8)	ND	ND	ND	ND
Coccolysin (EC 3.4.24.30)	31	18	5	4
Ficin (EC 3.4.22.3)	44	22	6	14
Ginger protease (EC 3.4.22.67)	ND	ND	ND	ND
Glutamyl endopeptidase II (EC 3.4.21.82)	ND	ND	ND	ND
Glycyl endopeptidase (EC 3.4.22.25)	1	2	ND	1
Leukocyte elastase (EC 3.4.21.37)	58	24	3	2
Metridin (EC 3.4.21.3)	20	14	2	9
Oligopeptidase B (EC 3.4.21.83)	3	2	ND	ND
Oligopeptidase F (-)	11	6	1	5
Pancreatic elastase (EC 3.4.21.36)	62	28	3	1
Pancreatic elastase II (EC 3.4.21.71)	16	8	1	6
Papain (EC 3.4.22.2)	60	25	4	13
Pepsin (pH > 2) (EC 3.4.23.1)	82	47	4	20
Plasmin (EC 3.4.21.7)	3	2	ND	ND
Prolyl oligopeptidase (EC 3.4.21.26)	ND	ND	ND	ND
Proteinase K (EC 3.4.21.67)	45	30	8	9
Proteinase P1 (EC 3.4.21.96)	29	21	2	8
Subtilisin (EC 3.4.21.62)	37	21	6	18
Thermolysin (EC 3.4.24.27)	40	25	7	2
Trombin (EC 3.4.21.5)	ND	ND	ND	ND
Trypsin (EC 3.4.21.4)	3	2	ND	ND
Trypsin + chymotrypsin A	46	26	8	14
Trypsin + chymotrypsin C	67	35	6	14
Pepsin + trypsin + chymotrypsin A	83	42	4	20
Pepsin + trypsin + chymotrypsin C	76	43	5	20
Xaa-Pro dipeptidase (EC 3.4.13.9)	ND	ND	ND	ND
V-8 protease (pH = 4) (EC 3.4.21.19)	ND	ND	ND	ND

* Other activities: antibacterial, immunomodulating, stimulating, neuropeptide, regulating, inhibitor, hypotensive, activating ubiquitin-mediated proteolysis, renin inhibitor, CAMPDE inhibitor, glucose uptake stimulating peptide. * ND: Not Detected.

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
