# Peer review of "In Silico Analysis of Bioactive Peptides Released from Giant Grouper (Epinephelus lanceolatus) Roe Proteins Identified by Proteomics Approach"

_molecules, 2018, doi:10.3390/molecules23112910_

Reviewer 1 Report

This is an interesting paper that presents an original and novel approach regarding the giant grouper roe as a potential raw material for the generation of pharmaceutical products.
However before it may be published, there are some issues that have to be addressed as provided below.
1. Indeed, ACE-inhibiting peptides are promising alternative in hypertension therapy  but it needs to be mentioned that they are efficient in the pre-hypertension treatment. In other stages of "more advanced" hypertension  food derived ACE inhibitors are supportive agents in the therapy when combined with drugs. Thus, I would suggest to modify the fragment of text concerning this issue.
2. Please use the name BIOPEP-UWM database instead of BIOPEP database.
3. Please, provide EC numbers of enzymes used in the simulated enzymatic hydrolysis (Table 3).

Author Response

Response to Reviewer 1 Comments

Point 1: Indeed, ACE-inhibiting peptides are promising alternative in hypertension therapy  but it needs to be mentioned that they are efficient in the pre-hypertension treatment. In other stages of "more advanced" hypertension  food derived ACE inhibitors are supportive agents in the therapy when combined with drugs. Thus, I would suggest to modify the fragment of text concerning this issue.

The explanation has been added in Line 44 to 48.

Point 2: Please use the name BIOPEP-UWM database instead of BIOPEP database.

“BIOPEP database” in line no.17, 20, 69, 72, 138, 141, 145, 152, 186, 235, 247, 249, 251, 256, 258, 360, 393 have been corrected into “BIOPEP-UWM database”.

Point 3: Please, provide EC numbers of enzymes used in the simulated enzymatic hydrolysis (Table 3).

The information about EC numbers of enzymes (Table 3) has been added.

Reviewer 2 Report

The subject of investigation is worthy for peptide discovery and use in the field of pharmaceutical biotechnology, however the present study is incipient.

The authors should explore the potentialities of combined MS and bioinformatic analysis, as well as should conduct some functional assays with the cleaved peptide from giant grouper's roe; for instance, to identify cleaved peptide after digesting grouper roe with one or two selected proteases. Apart of in gel digestion for peptide identification, it would be essential to conduct in solution digestion, since a number of protein is apparent in the gel that was not identified.

The article needs a complete reorganization and English proofreading.

It would be great if the could author re-submit an improved version of this manuscript.

Author Response

Response to Reviewer 2 Comments

Point 1: The authors should explore the potentialities of combined MS and bioinformatic analysis, as well as should conduct some functional assays with the cleaved peptide from giant grouper's roe; for instance, to identify cleaved peptide after digesting grouper roe with one or two selected proteases.

1)     The explanation about the combined MS and bioinformatics analysis is in Paragraph 3 (Introduction section). Authors also mentioned some previous studies using the proteomics technique coupled with bioinformatics tool (Line 69) and the advantage of these techniques (Line 76).

2)     This study was conducted as prediction analysis of giant grouper roe protein for in vitro analysis which is still in the progress in our laboratory.

Point 2: Apart of in gel digestion for peptide identification, it would be essential to conduct in solution digestion, since a number of protein is apparent in the gel that was not identified.

The reason using in-gel digestion is that authors targeted some protein bands based on molecular weight reported in literatures. Giant grouper roe protein was characterized using SDS-PAGE to observe protein band’s location or migration pattern where MW of each bands can be estimated using standard protein markers. In addition, the thickness of protein band indicates higher level of protein in a particular molecular weight. These considerations were then used to choose which band will be subjected to in-gel digestion.

Point 3: The article needs a complete reorganization and English proofreading. It would be great if the could author re-submit an improved version of this manuscript.

Some statement fragments were revised and reorganized accordingly and relevant information were supplemented. Please refer to the revised manuscript.

Reviewer 3 Report

This manuscript (molecules-380061) deals with potential bioactive peptides from roe proteins of giant grouper based on in silico analysis. This manuscript was interesting; however, some major concerns existed before drawing the conclusions. Moreover, some methods used were quite preliminary.

 In addition, for some reasons, I can not see the supplementary documents in the reviewer system. Have authors uploaded them?

Some majors concerns were summarized blow.

1. In terms of BIOPEP analysis, the data were obtained in March 2018, which was an old version. The authors should check the latest data prior to submission, as some latest peptide sequences with bioactivity can be updated. Therefore, the predicted results will be different. Therefore, authors should supplement the relevant information in the discussion section.

2. In Table 3, authors claimed that Pepsin (pH> 2) was the most promising enzyme to release high number of DPP-IV and ACE-I inhibitory fragments. However, in the real situation, the optimal pH for pepsin is 2. Authors must provide more information in this regard.

3. As for in silico hydrolysis by enzyme combinations, e.g.  Pepsin + trypsin + chymotrypsin, a key issue should be emphasized. Will these 3 enzymes cleave the protein sequence at the same time or in a sequential manner? Authors should add more discussions.

4. In addition, the major difference between in silico, in vitro and in vivo hydrolysis should be supplemented. Because authors indicated a number of bioactive fragments can be released from roe proteins of giant grouper. Neverthess, authors failed to verify whether they could be released in the practical and experimental digestions. Authors may find some useful discussion in the review paper entitled Exploration of collagen recovered from animal by-products as a precursor of bioactive peptides: Successes and challenges.

5. I can NOT find Fig. 2 in the main text.

6. What was the aim for BLAST analysis of vitellogenin sequences? Can similar protein sequence release similar bioactive fragments? Authors need to comment it.

7. I do not agree with the conclusion. The experimental digestions must be performed. Or else, authors should discuss the difference between in silico and in vitro digestions.

Author Response

Response to Reviewer 3 Comments

Point 1: This manuscript (molecules-380061) deals with potential bioactive peptides from roe proteins of giant grouper based on in silico analysis. This manuscript was interesting; however, some major concerns existed before drawing the conclusions. Moreover, some methods used were quite preliminary.

 In addition, for some reasons, I can not see the supplementary documents in the reviewer system. Have authors uploaded them?

Supplementary data has been added. Pls. refer to Line 390.

Point 2: In terms of BIOPEP analysis, the data were obtained in March 2018, which was an old version. The authors should check the latest data prior to submission, as some latest peptide sequences with bioactivity can be updated. Therefore, the predicted results will be different. Therefore, authors should supplement the relevant information in the discussion section.

BIOPEP-UWM database result has been updated on 30 October 2018. Some results have been revised based on the recent data obtained from BIOPEP-UWM database.

The relevant information has been added from Line 253 to 255.

Point 3: In Table 3, authors claimed that Pepsin (pH> 2) was the most promising enzyme to release high number of DPP-IV and ACE-I inhibitory fragments. However, in the real situation, the optimal pH for pepsin is 2. Authors must provide more information in this regard.

Table 3 has been updated to the recent data obtained from BIOPEP-UWM database (30 October 2018). The result showed that pepsin (pH > 2) was predicted to release better number of bioactivities from vitellogenin sequences.

Qiao et al. (2002) mentioned that the time needed to maximize pepsin utilization under in vitro analysis depended on pH and the concentration of pepsin where the pH ranged from 2 to 4. In another research done by Gray et al. (2014), pepsin is stable over the pH range of 1-6. They reported that pepsin has maximum activity at pH 2; however, 70% of the maxi-mal pectic activity is still present at pH 4.5, and almost no peptic activity at pH 5.5.

References:

Gray, V. A., Cole, E., Toma, J. M. R., Ghidorsi, L., Guo, J.-H., Han, J.-H., Han, F., Hosty, C. T., Kochling, J. D. & Kraemer, J. (2014). Use of enzymes in the dissolution testing of gelatin capsules and gelatin-coated tablets--revisions to Dissolution< 711> and Disintegration and Dissolution of Dietary Supplements< 2040. Dissolution Technologies, 21, 6-20.

Qiao, Y., Gumpertz, M. & Van Kempen, T. (2002). Stability of pepsin (EC 3.4. 23.1) during in vitro protein digestability assay. Journal of Food Biochemistry, 26, 355-375.

Point 4: As for in silico hydrolysis by enzyme combinations, e.g.  Pepsin + trypsin + chymotrypsin, a key issue should be emphasized. Will these 3 enzymes cleave the protein sequence at the same time or in a sequential manner? Authors should add more discussions.

Hydrolysis simulation in BIOPEP database does not mention about how the computational proteolysis is done. These three enzymes are chosen and run simultaneously in one proteolysis simulation.

Point 5: In addition, the major difference between in silico, in vitro and in vivo hydrolysis should be supplemented. Because authors indicated a number of bioactive fragments can be released from roe proteins of giant grouper. Neverthess, authors failed to verify whether they could be released in the practical and experimental digestions. Authors may find some useful discussion in the review paper entitled Exploration of collagen recovered from animal by-products as a precursor of bioactive peptides: Successes and challenges.

The explanation has been added in Line 263 to 276.

Point 6: I can NOT find Fig. 2 in the main text.

Correction has been made.

Point 7: What was the aim for BLAST analysis of vitellogenin sequences? Can similar protein sequence release similar bioactive fragments? Authors need to comment it.

BLAST analysis was applied to compare the two vitellogenin sequences from Epinephelus coioides (NCBI accession number: ACI01807.1) and Epinephelus lanceolatus (identified from another research) and to calculate regions of commonality of AA sequences within protein. The reason is that vitellogenin sequences detected in this research using Mascot database was from Epinephelus coioides, while authors used roe sample from Epinephelus lanceolatus. Therefore, Blast analysis was done to see the commonality of sequences between those protein sequences.

Based on in silico analysis, similar protein sequences could release similar bioactive fragments. Furthermore, according to quantitative structure-activity relationship (QSAR) study, bioactivities are mainly dependent on AA sequences. Therefore, similar protein sequences could release similar bioactive fragments. Nonetheless, some factor such as enzyme used in hydrolysis might give different results.

Point 8: I do not agree with the conclusion. The experimental digestions must be performed. Or else, authors should discuss the difference between in silico and in vitro digestions.

This study was conducted as prediction analysis for in vitro analysis which is still on process in our laboratory. As authors mentioned from Line 265, the result between in vitro and in silico analysis might be different due to some factors affected during hydrolysis.

Authors revised the statement “an effective and efficient method” into “a rapid method” in conclusion section. Pls. refer to line 374.

Round  2

Reviewer 2 Report

Although the revised version did not improved substantially the formal presentation of the manuscript, it is now acceptable to communicate the current research and data. The theme is interesting for the scientific community and a set of additional practical experiments should be realized in the future, in real in vitro conditions with the aim of analyzing the diversity of released peptides from the giant grouper roe.

Reviewer 3 Report

Authors have addressed my previous concerns.